# Eco-Friendly Polymer Nanocomposite Coatings for Next-Generation Fire Retardants for Building Materials

**DOI:** 10.3390/polym16142045

**Published:** 2024-07-17

**Authors:** Haradhan Kolya, Chun-Won Kang

**Affiliations:** Department of Housing Environmental Design, Research Institute of Human Ecology, College of Human Ecology, Jeonbuk National University, Jeonju 54896, Jeonbuk, Republic of Korea; hdk@jbnu.ac.kr

**Keywords:** carbon neutrality, wood, eco-friendly, polymer coatings, fire resistance

## Abstract

The increasing global commitment to carbon neutrality has propelled a heightened focus on sustainable construction materials, with wood emerging as pivotal due to its environmental benefits. This review explores the development and application of eco-friendly polymer nanocomposite coatings to enhance wood’s fire resistance, addressing a critical limitation in its widespread adoption. These nanocomposites demonstrate improved thermal stability and char formation properties by integrating nanoparticles, such as nano-clays, graphene oxide, and metal oxides, into biopolymer matrices. This significantly mitigates the flammability of wood substrates, creating a robust barrier against heat and oxygen. The review provides a comprehensive examination of these advanced coatings’ synthesis, characterization, and performance. By emphasizing recent innovations and outlining future research directions, this review underscores the potential of eco-friendly polymer nanocomposite coatings as next-generation fire retardants. This advancement supports the expanded utilization of wood in sustainable construction practices and aligns with global initiatives toward achieving carbon neutrality.

## 1. Introduction

The international community, including key organizations such as the United Nations Framework Convention on Climate Change (UNFCCC), has increasingly prioritized policies aimed at reducing carbon emissions to combat the climate crisis [1]. In this context, the issue of fire safety, particularly in structures made of wood, takes on a new urgency. One significant milestone in the global effort to combat climate change is the widespread declaration of carbon neutrality, or net-zero emissions, by around 130 countries, including the EU, the United States, and Japan, with some nations already enacting legislation to achieve this goal by 2050 [2,3]. Central to these initiatives is the promotion of green building materials, particularly wood, which is recognized for its renewability and environmental benefits [4,5]. However, the inherent combustibility of wood poses significant challenges in ensuring the safety and resilience of structures against fire hazards [6]. Traditional fire retardants, often based on halogenated compounds, have been effective in enhancing fire resistance but pose significant environmental and health risks [7]. These conventional fire retardants release toxic gases and persistent organic pollutants during combustion, leading to severe environmental and human health issues [8]. Consequently, there is a pressing need to develop safer, more sustainable alternatives that can provide effective fire protection without compromising environmental integrity [9].

Considering the pressing need for safer fire retardants, the focus has shifted to eco-friendly alternatives. The application methods for these retardants on wood-based materials include impregnation [10] and surface coatings [11]. Impregnation, although effective in penetrating deep into the wood, is often expensive and leads to substantial chemical and solvent wastage, raising environmental and economic concerns. In contrast, coatings offer a more efficient and less wasteful alternative, especially when using adhesive polymers to ensure durability and effectiveness. These eco-friendly coatings not only enhance fire resistance but also contribute to a more sustainable and environmentally conscious approach to fire safety.

Recently, eco-friendly polymer nanocomposite coatings have garnered significant attention from researchers [12]. Bio-composites made from renewable polymers, such as poly (lactic acid) [13], furfuryl alcohol [14], gluten [15], soy flour [15], and starch [16], combined with naturally available fibers, have been attracting significant interest due to their environmentally friendly characteristics [12]. These coatings incorporate nanoparticles, such as graphene [17], layered double hydroxides (LDH) [18], natural clay minerals [19], MXene [20], nano-metal oxides [21], polyphosphazene [22], cellulose nanofibrils (CNF) [23], biochar [24], and responsive color-changing materials. Notably, graphene oxide (GO), a two-dimensional carbon-based nanomaterial [25], and chitin composites have garnered significant attention in recent decades [26]. Natural clay minerals, including montmorillonite, kaolin, and bentonite, are frequently utilized in the composites industry due to their widespread availability, recyclability, and non-toxic nature [27]. The resulting nanocomposites provide enhanced fire resistance and align with green chemistry principles due to their low toxicity and biodegradability. Therefore, a review focusing on the development of novel eco-friendly polymer nanocomposite coatings for fire retardancy could provide valuable insights for researchers.

This review provides an overview of the current state of eco-friendly polymer nanocomposite coatings for fire retardancy. It examines fire-retardant treatment methods, including impregnation and coatings, and explores various nanofillers, such as layered silicates, carbon-based nanomaterials, and bio-based nanoparticles. The review covers these nanocomposites’ synthesis, fabrication, and characterization and the mechanisms underlying their fire-retardant properties. Environmental and health considerations, recent advances, case studies, and prospects are also discussed. This comprehensive analysis offers valuable insights for researchers and industry professionals in developing safer and more sustainable fire-retardant systems.

## 2. Fire-Retardant Chemicals

Fire-retardant chemicals are substances used to reduce the flammability of materials and delay their combustion. Traditional fire retardants include halogenated compounds, ammonium polyphosphate, borates, and intumescent systems [28]. Halogenated compounds, such as brominated and chlorinated flame retardants, are highly effective but pose significant environmental and health risks due to releasing toxic gases during combustion [8,29]. Ammonium polyphosphate and borates work by promoting char formation and suppressing flame propagation, making them safer alternatives [30]. Intumescent systems form a protective char layer during heat exposure, providing an insulating barrier that slows down combustion [31]. Despite their effectiveness, many conventional fire retardants face scrutiny for their potential toxicity and environmental persistence, leading to a growing interest in developing eco-friendly alternatives [32]. A list of fire-retardant chemicals with their chemical formulas and melting points (MP) or boiling points (BP) are shown in Table 1.

### 2.1. Treatment Methods

Fire retardants can be applied to materials using several methods, such as impregnation and coating, each with its distinct advantages and drawbacks.

#### 2.1.1. Impregnation

The impregnation process for fire retardants involves several meticulous steps to ensure deep penetration and effectiveness in enhancing the fire resistance of wood. First, the wood is prepared by drying it to a specific moisture content, as the moisture level affects the uptake of the fire-retardant solution. Second, fire-retardant chemicals are dissolved in suitable solvents to create a homogeneous solution. Common fire retardants include ammonium polyphosphate, borates, and other inorganic salts, with the choice of solvent depending on these chemicals’ solubility and ability to penetrate the wood structure effectively. Water is the most common solvent, but organic solvents can also be used for specific formulations. Third, the wood is placed in a vacuum chamber, and a vacuum is applied to remove air from the wood’s pores, facilitating deeper penetration of the fire-retardant solution. Fourth, after the vacuum treatment, the fire-retardant solution is introduced into the chamber, and the pressure is increased to force the solution into the wood’s cellular structure. This pressure treatment ensures the chemicals reach deep into the wood, providing comprehensive fire protection. Fifth, the impregnated wood is removed from the chamber and dried to remove excess solvent and stabilize the wood, which can be performed using kilns or air drying, depending on the desired final moisture content and the type of wood. A schematic of the vacuum pressure impregnation chamber is shown in Figure 1a.

The impregnation process for fire retardants is a highly effective method to enhance the fire resistance of wood by introducing protective chemicals deep into its structure [59]. This method involves sophisticated equipment and precise control over various factors, including solvent selection, chemical solubility, and wood pore structure. Impregnation with fire-retardant chemicals can lead to color changes in the wood due to the interaction between the chemicals and the natural components of the wood, such as tannins and lignin [60,61]. Depending on the fire-retardant formulation, the treated timber may darken or take on a different hue. While this color change can sometimes be mitigated through post-treatment processes or additives that stabilize the wood’s appearance, it is often an unavoidable aspect of chemical impregnation. Throughout this process, considerations such as the solubility of chemicals, dispersion within the wood’s cell wall, and the wood’s pore structure are paramount, influencing the efficacy and uniformity of the treatment. Impregnation, although effective in penetrating deep into the wood, is often expensive and leads to substantial chemical and solvent wastage, raising environmental and economic concerns [62].

#### 2.1.2. Coatings

Coating methods offer an alternative approach to applying fire retardants to materials, including wood, offering distinct advantages over impregnation [63]. Below, we delve into the details of coating methods for fire retardants, exploring the procedures, types of coatings, application techniques, and considerations involved. Prior to coating, meticulous surface preparation is imperative. This involves cleaning and smoothing the material’s surface to ensure optimal adhesion and uniform coverage of the fire-retardant coating (as shown in Figure 1b). The formulation process entails blending fire-retardant chemicals with appropriate binders and additives to create a homogeneous mixture. Standard binders encompass adhesive polymers, such as acrylics, epoxies, and polyurethanes. A list of polymers with their chemical formulas and melting or boiling points is noted in Table 2.

Different coating processes include spraying, brushing, rolling, dipping, and electrostatic coating. After application, the coated material can cure or dry, depending on the type of coating used [87]. Depending on the coating formulation, curing may involve air-drying, heat-curing in ovens, or exposure to ultraviolet (UV) light. Coatings offer a cost-effective, flexible, and environmentally friendly alternative to impregnation, though they may require more frequent maintenance and provide primarily surface-level protection.

## 3. Synthesis and Characterization of Fire-Retardant Polymers

Historical records show the use of various materials to develop flame-retardant properties. Over time, alum, ferrous sulfate, stannic oxide, borax, and ammonium phosphates were used to absorb heat and prevent fire spread. The infusion of financial resources and advancements in polymeric materials has led to the introduction of hybrid materials for fire safety. Generally, solution-mixing, in situ, and ex situ methods are employed to create flame-retardant polymer nanocomposites by optimizing the composition and conditions. In the in situ method, nanomaterials are dispersed in a liquid monomer, with surfactants controlling filler agglomeration and geometry during polymerization. This method enhances porosity, making the composite lighter.

The solution-mixing method involves dissolving one or more components in a solvent to create a solution, which is then mixed thoroughly to ensure uniform distribution of the components. After mixing, the solvent may be evaporated or removed to yield the final product. The ex situ methods often involve preparing or modifying materials outside their intended operational environment. For example, nanoparticles can be synthesized separately and incorporated into a composite material. A schematic of each process is shown in Figure 2.

Generally, the characterization of prepared materials involves thermal analysis (DSC, DTA, TGA, TMA, and DMA) [90], microscopy (TEM, SEM, and AFM) [91], spectroscopy (UV-Visible, FTIR, NMR, and Raman) [92], tribological properties [93], and X-ray diffraction techniques, which are employed for chemical characterization of polymer nanocomposites [94]. Mechanical properties for structural stability are assessed using universal testing machines, dynamic mechanical analyzers, and impact and surface analyzers [95]. Weathering effects and electrical properties are measured with resistivity meters and dielectric strength analyzers [96]. Chakraborty et al. provided detailed insights into microscopy and analytical techniques for cellulose morphological, structural, chemical, and thermal characterization [97].

Flame retardancy assessment for fire-retardant polymer nanocomposites uses cone calorimetry and UL-94 tests [98,99]. UL-94, a standard preliminary test (as shown in Figure 3a), categorizes polymers based on vertical and horizontal burn testing and thin films, grading them into V_0_, V_1_, and V_2_ based on the burning time, rate, and dripping behavior (details are shown in Table 3) [100]. Limiting of oxygen index (LOI) testing (ISO 4589-2 [101]) measures the minimum oxygen concentration needed to sustain combustion in a closed atmosphere, with better flame-retardant materials requiring higher oxygen concentrations (as shown in Figure 3b).

Cone calorimetry is a highly effective test for evaluating the fire behavior of medium-sized polymer samples. This method measures the decrease in oxygen concentration in the combustion gases of a sample exposed to a specific heat flux, generally between 10 and 100 kW/m^2^. In the United States, it is standardized under ASTM E 1354 [102] and covered by the international standard ISO 5660 [103]. In this test, a sample measuring 100 × 100 × 4 mm^3^ is placed on a load cell to monitor mass loss throughout the experiment. The sample is irradiated uniformly from above by a conical radiant electric heater, and combustion is initiated using an electric spark (Figure 3c). The resulting combustion gases pass through the heating cone and are captured by an exhaust system equipped with a centrifugal fan and hood. This system measures gas flow, oxygen, CO, CO_2_ concentrations, and smoke density.

The data from gas flow and oxygen concentration are used to calculate the heat release rate (HRR), expressed in kW/m^2^, indicating the amount of heat released per unit time and surface area. The progression of HRR over time, especially its peak value (pHRR or HRRmax), is critical for assessing fire properties. The total heat release (THR), expressed in kJ/m^2^, is obtained by integrating the HRR over time. Additionally, this test provides information on the time to ignition (TTI), duration of combustion or extinction (TOF), mass loss during combustion, quantities of CO and CO_2_ produced, and total smoke released (TSR), as depicted in Figure 3d [100].

## 4. Mechanism of Flame Retardancy

The flame retardancy mechanisms of fire-retardant polymer nanocomposites involve gas phase inhibition, char formation or heat sink effect, and cooling through the generation of insulating layers. Gas phase inhibition occurs when flame retardants added to gases produced during polymer heating trap free radicals, halting the combustion process, typically seen with halogenated flame retardants (Figure 4a) [104]. Char formation involves flame-retardant chemicals reacting with the material’s surface to create a carbonaceous layer, insulating the polymer, and reducing pyrolysis and gas release during burning, often used with non-halogen systems utilizing phosphorous and nitrogen chemistries [105]. The formation of carbonaceous char reduces the release of volatile by-products. The mechanism depicting the reactivity of phosphorus-based flame retardants in the gas phase is shown in this paper [26]. The cooling mechanism involves endothermic reactions releasing water molecules, which cool the polymer and dilute combustion, with hydrated metal salts, such as aluminum trihydroxide, utilized for this purpose. Additionally, synergistic approaches combining different flame retardants, such as antimony oxide, with halogen-containing ones, enhance flame retardancy by inhibiting vapor phase combustibility and generating heat sink behavior in polymer composites (Figure 4b) [105].

## 5. Research on Flame-Retardant Chemicals

Research on flame-retardant treatment and combustion characteristics of wood has been evolving over the years, with studies from different years providing valuable insights into the effectiveness of various flame-retardant formulations and treatment methods. For instance, Park et al. [107] conducted a thermal analysis to examine the combustion characteristics of fire-retardant-treated wood. Their findings indicated that flame-retardant treatment significantly influences wood’s thermal decomposition and combustion properties. Zhang Zhi-jun et al. [108] conducted a fire retardation performance test on a wood flour/polystyrene composite (WF-PS) treated with ammonium polyphosphate (APP) and tested using a cone calorimeter. The study found that the heat release rate was 35 kW/m^2^, significantly reducing the total heat release. Additionally, the treatment with APP extended the ignition time of the composite, indicating improved fire-retardant properties.

Jinxue Jiang et al. [109] found that flame-retardant-treated wood exhibited the highest limiting oxygen index (LOI) values, indicating synergistic interactions between phosphorus and nitrogen (P–N). As the degree of degradation increased, the activation energies of the treated wood decreased by 19.6–50.4% compared to untreated wood. This led to higher char formation and reduced production of combustible products during degradation. These findings highlight the effectiveness of the P–N flame-retardant treatment in enhancing the fire resistance of wood by promoting char formation and inhibiting combustion. Lin Zhou et al. [110] investigated the effects of ammonium polyphosphate (APP) and 3-(methylacryloxyl) propyltrimethoxy silane-modified APP (M-APP) on wood flour/polypropylene composites (WF/PP). M-APP significantly improved the mechanical properties of WF/PP composites and acted as an effective flame retardant, surpassing the performance of APP according to cone calorimetry results. Moreover, M-APP enhanced char formation, as evidenced by SEM analysis, indicating its potential in enhancing the fire resistance of WF/PP composites through improved char formation capabilities. Using cone calorimetry and thermogravimetric analysis (TGA), the study found significant reductions in the peak heat release rate (HRR) by 21% and total heat release (THR) by 44.2%.

Dong Won Son et al. [111] treated Japanese red pine, hemlock, and radiate pine with inorganic chemicals, such as 50% sodium silicate, 3% boric acid, 3% ammonium phosphate, and 3% ammonium borate, using a vacuum/pressure (vacuum 78 kPa, 30 min, pressure 18 kg/cm^2^, time 2 h) impregnated method. The study reported that the ignition time of the treated wood was effectively delayed by these treatments, particularly with sodium silicate, ammonium phosphate, and ammonium borate, demonstrating an improvement in fire resistance. Seo et al. [112] analyzed the combustion and thermal properties of wood used indoors, such as the heat release rate, total heat release, and gas generation, using TGA and a cone calorimeter (KS F ISO 5660-1 [103]). They found that wood’s material properties significantly impacted its combustion behavior, and the formation of a carbonization layer notably varied by tree species, showing a high correlation between total heat release and weight loss. The study emphasized that ignition time and total heat release are crucial data for imparting fire resistance performance to wood.

Chai, et al. [113] evaluated the fire retardation effect on Cryptomeria fortunei wood treated with a boric acid-urea-formaldehyde (MUF) resin mixed with borax. The treatment increased the oxygen index and time to ignition (TTI). Park et al. [114] compared the flame-retardant performance and combustion characteristics of cypress wood and particle board. Cypress wood injected with flame-retardant resin via vacuum pressurization outperformed specimens treated with surface flame-retardant paint. In their study, this group demonstrated that water-soluble phosphate flame retardants, when mixed with poly ammonium phosphate, guanylurea phosphate, phosphoric acid, and resin, effectively impregnated perforated Hinkoi plywood used as a sound absorber. They found that the frequency of perforations influenced the impregnation process, with narrower hole spacing leading to increased impregnation. This resulted in a 15% improvement in flame-retardant performance compared to untreated samples [115].

Sathasivam Pratheep Kumar et al. [116] applied a composite coating of sodium silicate and clay minerals to wood as an innovative inorganic flame retardant. Cone calorimeter tests revealed that the composite-coated wood significantly reduced the heat release rate, delayed total heat release and ignition, and exhibited superior flame retardation compared to other tested coatings due to the dense surface layer. The addition of vermiculite to sodium silicate enhanced ignition resistance and resulted in lower HRR values than the wood coated solely with sodium silicate. Ribeiro et al. [117] developed an unsaturated polyester-based composite with enhanced fire retardancy using nano/micro-oxide particles and common flame retardants. Results showed that hybrid-flame-retardant systems improved fire properties but sometimes decreased mechanical properties due to poor matrix-filler adhesion.

Rocha et al. [118] engineered a high-density polyethylene composite reinforced with lignocellulosic fibers as a potential substitute for natural pine wood. The HDPE/sponge gourd fiber composite demonstrated the best impact resistance, and with the addition of magnesium hydroxide, it showed improved flammability and thermal stability. Sheng Li et al. [119] developed a biomass-based flame-retardant additive derived from renewable chitosan, melamine formaldehyde resin-coated ammonium polyphosphate, and organic montmorillonite, incorporated into waterborne epoxy resin (WBEP) for wood coatings. The resulting coated wood composites achieved a UL-94 V-0 rating, a limiting oxygen index of 31.8%, and maintained excellent flame-retardant performance even after water resistance tests. The biomass-based additive enhanced the carbonization capability, with residual char reaching 23.9 wt.% at 800 °C. Cone calorimeter tests showed reduced heat and smoke release, forming an effective char layer that protected the wood substrate. The WBEP coating demonstrated superior water resistance and flame-retardant efficiency, making it suitable for flame-retardant wood composites.

Lu et al. [61] impregnated melamine (MEL) with organic phosphoric acid (AP) into the porous structure of wood. The limiting oxygen index (LOI) and cone calorimetry tests showed that MEL/AP presence significantly improved fire resistance. The LOI value increased from 21.0% to 68.5%, and the peak heat release rate and total heat release amount decreased by 41.7% and 80.2%, respectively, compared to the control sample (as shown in Figure 5). This indicates that MEL/AP in a porous wood structure improves flame retardancy. Price et al. [120] developed tannic acid (TA)-based composites for fire safety, comparing them to pentaerythritol (PER) composites. TA composites significantly extended the time to failure from seconds or minutes to over 15 min, with a maximum of 27 min. They exhibited better fire performance, with lower peak heat release values (211 vs. 108 kW/m^2^), lower total heat release values (37.2 vs. 24.4 MJ/m^2^; as shown in Figure 6), and slower fire growth rates (2.43 vs. 1.27 kW/m^2^s^−1^).

X-ray photoelectron spectroscopy showed that TA char was more carbonaceous (54.71 at.% C vs. 39.63 at.% C in PER char). These findings demonstrate that TA composites provide superior fire protection, offering significant advancements for fire safety applications [120].

Özkan et al. [121] treated with fire retardants, including di-ammonium phosphate (DAP), borax, boric acid, and glucose, in aqueous solutions of 10%, 20%, and 30%. Post-heat treatment at 120 °C, 150 °C, and 180 °C improved water resistance, dimensional stability, mechanical strength, and thermal properties. The DAP/glucose complex enhanced phosphorus fixation, reducing leaching and providing long-term fire protection. These findings suggest potential for using treated wood in structural applications, offering enhanced durability and fire safety.

Yutao Yan et al. [122] developed a durable flame-retardant coating on wood using a layer-by-layer self-assembly method with chitosan (CS), graphene oxide (GO), and ammonium polyphosphate (APP; as shown in Figure 7). The coating enhanced thermal stability by lowering decomposition temperatures and increasing char residue due to the effective heat barrier properties of GO. Fire resistance significantly improved, with the LOI increasing from 22 to 42 and the heat release rate decreasing from 105.50 kW/m^2^ to 57.51 kW/m^2^ after 15 layers of CS-GO-APP were applied. The coating showed excellent durability in immersion and abrasion tests, forming a protective char layer that inhibited flame spread on wood surfaces.

Recently, Rantuch et al. [24] treated spruce wood with a furfuryl alcohol solution enriched with biochar via vacuum infiltration. The research aimed to assess the suitability of this treatment and evaluate its impact on thermal degradation properties. Thermal gravimetric analysis revealed that the biochar-furfurylated wood bio-composite (BFW) exhibited enhanced thermal stability compared to untreated wood (W) and furfurylated wood (FW). BFW also demonstrated improved fire characteristics, including decreased effective heat of combustion and carbon monoxide yield, highlighting its potential for enhancing wood’s fire resistance properties [24]. Besides, mineral fillers, such as aluminum and magnesium hydroxide, and natural mixtures, such as huntite and hydro-magnesite, are increasingly used as eco-friendly fire retardants. They act through endothermic decomposition, increasing the heat capacity of polymer residues, and enhancing the gas phase heat capacity with water or carbon dioxide. Despite the complexities in application across polymers, these fillers reduce flammability by up to 70%, as evidenced by LOI, UL-94, and cone calorimeter tests. Quantifying their effects reveals their nuanced impacts, crucial for sustainable fire safety solutions [123].

Liu et al. [124] explored the use of industrial lignin modified with phosphorus, nitrogen, and copper as a bio-based flame-retardant additive for wood-plastic composites (WPCs). The modified lignin (F-lignin) significantly enhanced the thermal stability and flame retardancy of WPCs compared to unmodified lignin (O-lignin). It reduced the heat release rate, total heat release, and smoke production during combustion, while promoting the formation of a dense, protective char layer. This innovative approach demonstrates the potential of utilizing industrial lignin in green flame-retardant strategies for sustainable WPC applications.

Moreover, Yang et al. [125] developed high-performance bio-composite materials from recyclable forestry waste, using lignin and cellulose as a natural adhesive matrix. Pretreatment with hydrogen peroxide, sodium hydroxide, and sodium silicate enhanced the material properties significantly, increasing tensile and bending strengths by over 90%. The bio-composite exhibited hydrophobicity with a water contact angle of 99.96° and maintained thermal stability up to 1300 °C without disintegration. These attributes make it highly suitable for eco-friendly construction applications, offering sustainable alternatives to petroleum-based materials. Yu et al. [126] introduced an eco-friendly method for creating flame-retardant wood composites using carboxymethylated alkali lignin, phytic acid, and melamine-urea-glyoxal resin. The resulting modified wood (MW/MPUC) showed significant improvements in flame retardancy, with a 56.8% reduction in total heat release, a 92.3% decrease in total smoke production, and an increase in the limiting oxygen index from 23.6% to 41.5% (as shown in Figure 8). Additionally, all modified wood samples passed the UL-94 V-1 flammability test and exhibited enhanced mechanical properties and dimensional stability.

From the above literature study, a summary table focusing on eco-friendly flame retardants is shown in Table 4.

### 5.1. Discussions

The trend in fire-retardant research has been shifting significantly toward eco-friendly polymer nanocomposite coatings, reflecting an increasing emphasis on sustainability and environmental safety. Early studies focused primarily on inorganic chemicals and conventional flame retardants, such as ammonium polyphosphate (APP), boric acid, and sodium silicate [108,109,110,111]. However, the growing awareness of the environmental impact and the potential health risks associated with these traditional retardants have driven researchers to explore green alternatives [7]. Recent advancements have seen the development of bio-based and eco-friendly flame retardants, utilizing natural materials such as chitosan, lignin, and cellulose, often enhanced with nanotechnology [122,123]. For instance, the use of carboxymethylated alkali lignin, phytic acid, and melamine-urea-glyoxal resin to create flame-retardant wood composites exemplifies this shift, achieving significant improvements in fire resistance and reduced smoke production while maintaining mechanical properties and dimensional stability [125]. Nanocomposite coatings incorporating materials such as graphene oxide, clay minerals, and biochar are becoming prominent, leveraging their superior thermal stability and char-forming capabilities to enhance fire retardancy [126]. These innovations not only meet the stringent fire safety standards but also align with the principles of green chemistry, paving the way for sustainable applications in construction and material science. This eco-conscious direction marks a significant evolution in the field, aiming to mitigate the environmental impact while ensuring effective fire protection.

### 5.2. Limitation, Challengs, and Opportunities

The shift toward eco-friendly polymer nanocomposite coatings for flame retardancy presents challenges and opportunities in the field. Challenges include higher production costs and the need to scale manufacturing processes to meet industrial demands. There are also concerns about potential compromises in mechanical properties and the long-term durability of these coatings under various environmental conditions. Standardizing testing protocols and gaining acceptance from industries and consumers are additional hurdles. However, these challenges are accompanied by significant opportunities. Advances in nanotechnology offer the potential to develop highly effective flame retardants with minimal environmental impact. Integrating renewable resources, such as chitosan, lignin, and cellulose, supports sustainability goals and enhances the value of agricultural by-products. Innovation in multifunctional coatings that offer additional benefits beyond fire resistance is another promising avenue. Moreover, increasing regulatory pressures and consumer demand for greener products create a favorable market environment for developing and adopting eco-friendly flame-retardant technologies. A probable future roadmap for eco-friendly flame-retardant materials is illustrated in Figure 9.

## 6. Conclusions

The evolution toward eco-friendly polymer nanocomposite coatings represents a pivotal advancement in fire-retardant technology for building materials. These coatings offer substantial opportunities, while facing challenges such as production costs, scalability, and ensuring mechanical integrity. Advances in nanotechnology enable the development of highly efficient flame retardants using renewable resources, aligning with sustainability objectives and regulatory requirements. Innovations in multifunctional coatings enhance fire resistance and provide added benefits, such as antimicrobial properties and thermal insulation. With growing market demand and regulatory support for greener solutions, the future holds promising prospects for integrating eco-friendly polymer nanocomposite coatings into mainstream building materials, ensuring safer environments while minimizing the environmental impact.

## Figures and Tables

**Figure 1 polymers-16-02045-f001:**
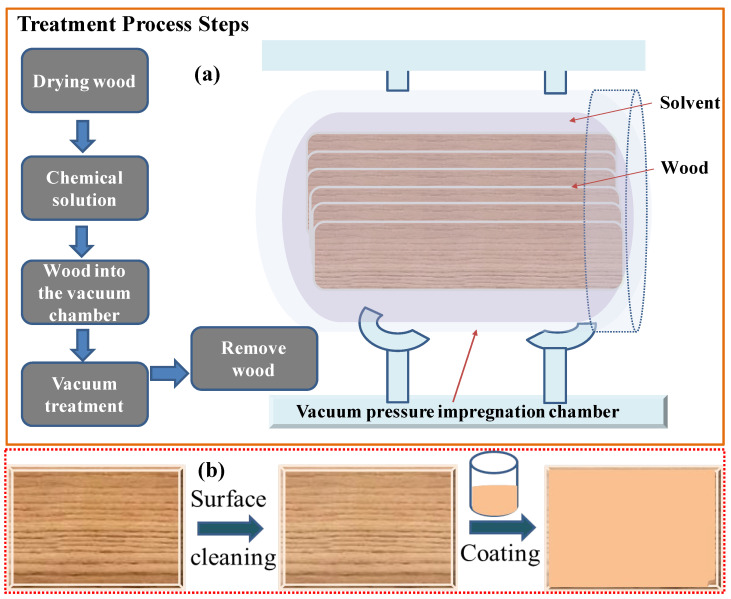
Schematic of (**a**) the vacuum pressure impregnation process and (**b**) surface coating.

**Figure 2 polymers-16-02045-f002:**
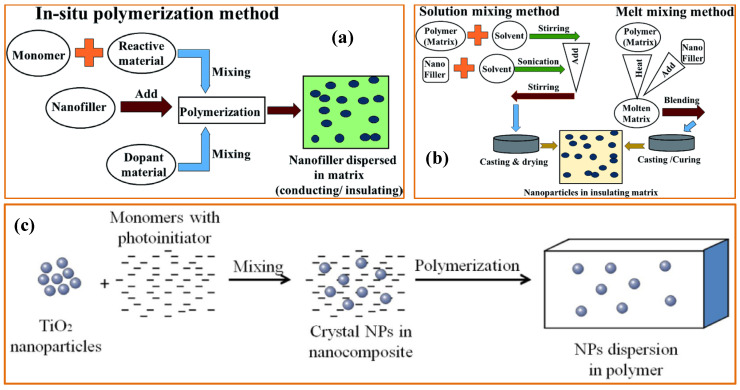
A schematic illustration of the preparation of fire-retardant polymer nanocomposites using (**a**) the in situ method and (**b**) the solution-mixing method [88]. Copyright 2019, reproduced with permission from the authors, RSC. (**c**) Ex situ method [89]. Copyright 2014, reproduced with permission from the authors, MDPI, Basel.

**Figure 3 polymers-16-02045-f003:**
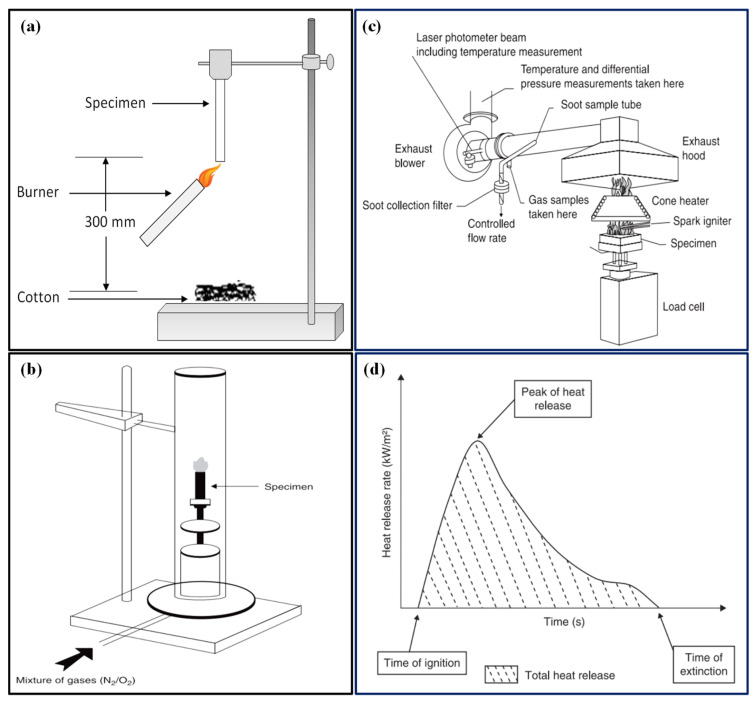
Schematic diagram of (**a**) the UL-94 vertical burning test [98]. Copyright 2020, reproduced with permission from the authors, MDPI, Basel. (**b**) The cone calorimeter, (**c**) the limiting oxygen index test, and (**d**) a typical cone calorimeter curve [100]. Copyright 2011, reproduced with permission from Woodhead Publishing Limited.

**Figure 4 polymers-16-02045-f004:**
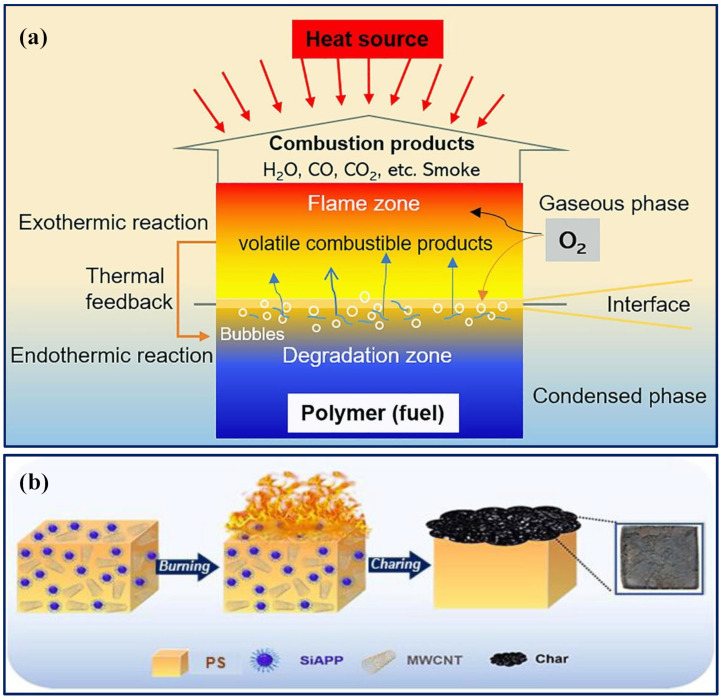
(**a**) The typical combustion process of polymers [104]. Copyright 2020, reproduced with permission from Published by Elsevier Ltd. (**b**) A schematic illustration of the flame-retardant nature [106]. Copyright 2023, reproduced with permission from Wiley-VCH GmbH.

**Figure 5 polymers-16-02045-f005:**
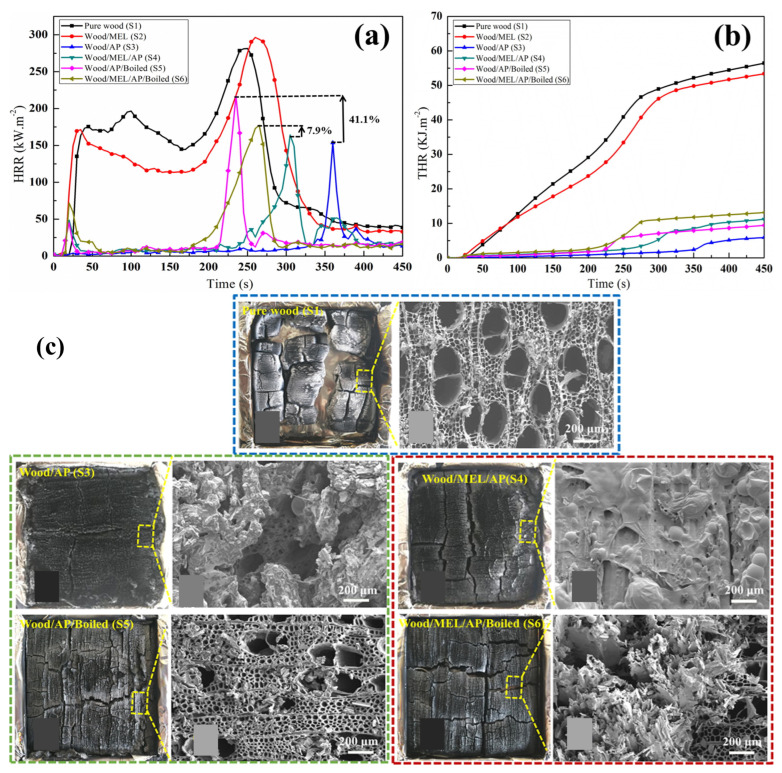
(**a**) Heat release rate (HRR) and (**b**) total heat release (THR) curves for both pure and modified wood. (**c**) Digital photographs and SEM images of pure wood and modified wood after cone calorimeter testing (CCT) [61]. Copyright 2020, reproduced with permission from Elsevier Ltd.

**Figure 6 polymers-16-02045-f006:**
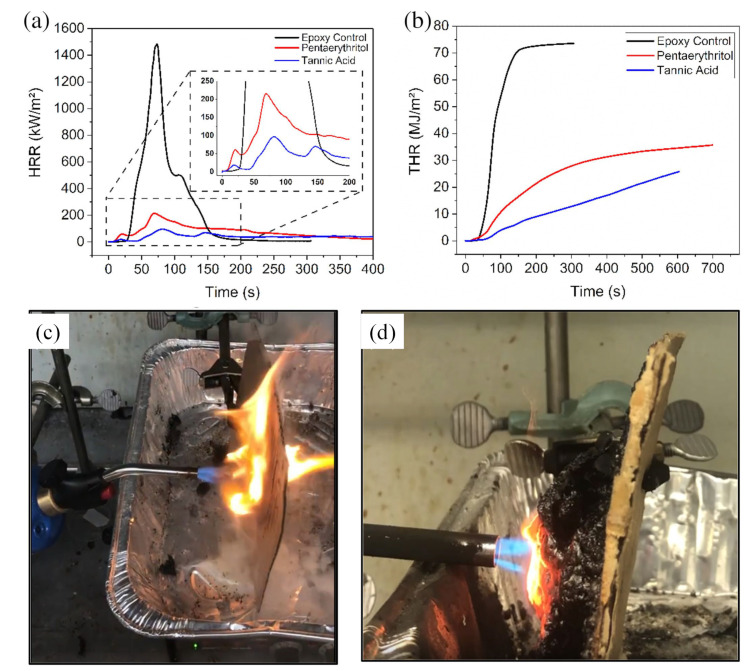
Cone calorimetry data illustrating (**a**) heat release rate (HRR) versus time and (**b**) total heat release (THR) versus time for cardboard coatings exposed to a propane blow torch. The uncoated cardboard was compromised after 7 s (**c**) and the 1 mm coated cardboard burned for 27 min (**d**). The coated sample exhibited a final expansion 25 times its original thickness [120]. Copyright 2020, reproduced with permission from the authors. SPE Polymers published by Wiley Periodicals LLC on behalf of the Society of Plastics Engineers.

**Figure 7 polymers-16-02045-f007:**
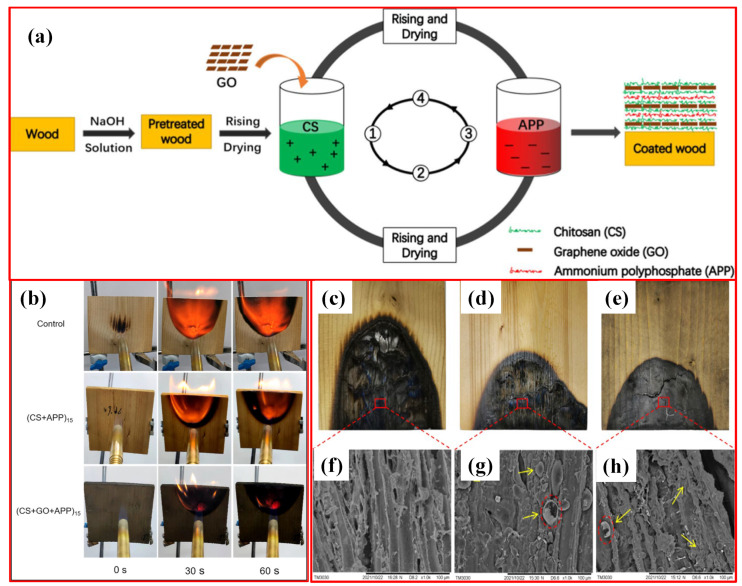
(**a**) Schematic illustration of the layer-by-layer (LBL) self-assembly CS-GO-APP coating on wood. (**b**) Fire behavior comparison of coated and uncoated wood samples at various burning times. Digital photographs and SEM images after combustion test: (**c**,**f**) uncoated wood, (**d**,**g**) (CS-APP)15-coated wood, and (**e**,**h**) (CS-GO-APP)15-coated wood [122]. Copyright 2022, reproduced with permission from the authors. Published by the American Chemical Society.

**Figure 8 polymers-16-02045-f008:**
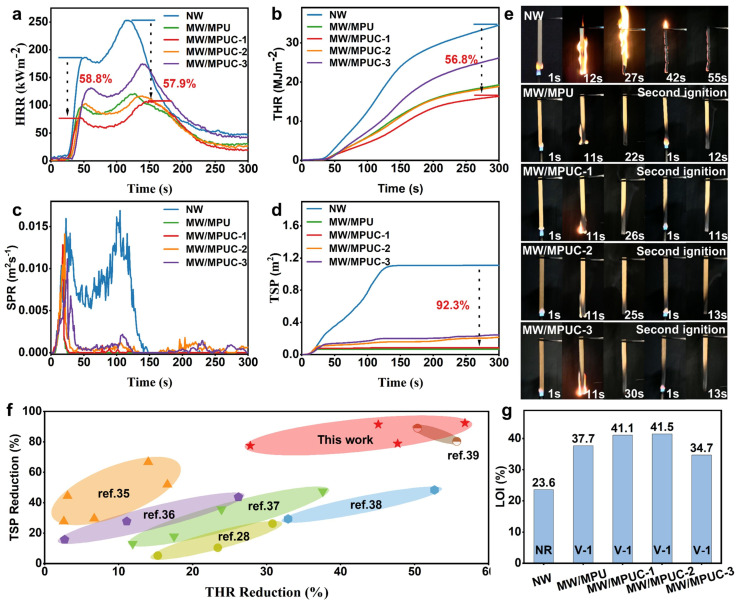
(**a**) The HRR curves, (**b**) the THR curves, (**c**) the SPR curves, and (**d**) the TSP curves of the specimens. (**e**) Digital photographs of specimens during UL-94 testing. (**f**) Summary of flame-retardant efficiency of PA and lignin, and (**g**) LOI and UL-94 testing results of the specimens [126]. Copyright 2024, reproduced with permission from Elsevier B.V. All rights are reserved.

**Figure 9 polymers-16-02045-f009:**
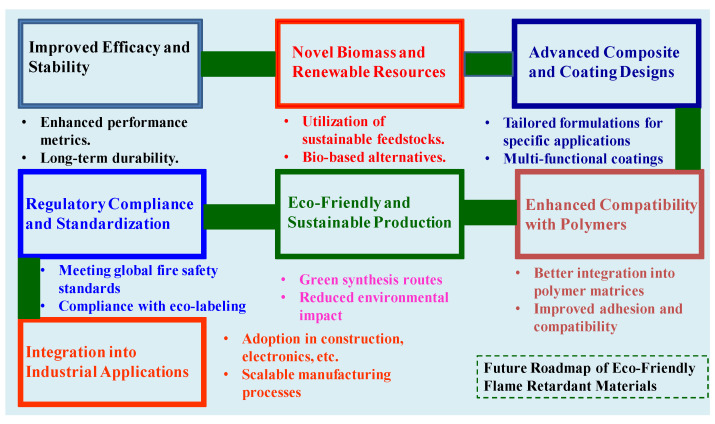
A probable future roadmap for the development and application of eco-friendly flame-retardant materials.

**Table 1 polymers-16-02045-t001:** List of fire-retardant chemicals, along with their chemical formulas and melting or boiling points (MP/BP), sourced from the internet (Wikipedia).

Fire-Retardant Chemicals	Chemical Formula	MP/BP (°C)	Ref.
Aluminum hydroxide	Al(OH)_3_	300	[33]
Aluminum phosphate	AlPO_4_	1800	[34]
Ammonium polyphosphate	(NH_4_)_3_PO_4_	260–320	[35]
Antimony trioxide	Sb_2_O_3_	656	[36]
Borax	Na_2_B_4_O_7_·10H_2_O	743	[37]
Boric acid	H_3_BO_3_	170	[38]
Brominated flame retardants	Varies	280–345	[39]
Chlorinated flame retardants	Varies	Varies	[39]
Diethyl ethyl phosphonate	C_6_H_15_O_3_P	198	[40]
Dimethyl methyl phosphonate	C_3_H_9_O_3_P	181	[41]
Graphene and graphene oxide	C, C_x_H_y_O_z_	3000	[42]
Graphite	C	3650	[43]
Hydrated lime	Ca(OH)_2_	580	[44]
Layered double hydroxides	Varies	>900	[45]
Magnesium hydroxide	Mg(OH)_2_	350	[46]
Melamine	C_3_H_6_N_6_	345	[46]
Melamine cyanurate	C_6_H_9_N_9_O_3_	350	[47]
Melamine phosphate	C_3_H_9_N_6_O_4_P	120–122	[48]
Melamine polyphosphate	C_3_H_9_N_6_O_4_P	>350	[49]
Red phosphorus	P_4_	590	[50]
Silicon dioxide (silica)	SiO_2_	1610	[51]
Trimethylphosphate	C_3_H_9_O_4_P	156	[52]
Triphenyl phosphate	C_18_H_15_O_4_P	50	[53]
Tris(1,3-dichloro-2-propyl) phosphate	C_9_H_15_C_l3_O_4_P	315	[54]
Zinc borate	Zn_2_B_6_O_11_·3.5H_2_O	980	[55]
Zinc oxide	ZnO	1975	[56]
Vanadium oxide	V_2_O_5_	1967	[57]
Tin oxide	SnO, SnO_2_	1630	[58]

**Table 2 polymers-16-02045-t002:** List of polymers used for making fire-retardant composite polymers, along with their melting or boiling points (MP/BP), sourced from the internet (Wikipedia).

Polymer or Adhesive Polymer	Chemical Formula	MP/BP (°C)	Ref.
Polyurethane (PU)	(C_3_H_8_N_2_O)_n_	136	[64]
Acrylic resins	(C_5_H_8_O_2_)_n_	160	[65]
Epoxy resins	(C_21_H_25_ClO_5_)n	120	[66]
Polyvinyl chloride (PVC)	(C_2_H_3_Cl)_n_	100	[67]
Polyethylene (PE)	(C_2_H_4_)_n_	115–135	[68]
Polypropylene (PP)	(C_3_H_6_)_n_	130–170	[69]
Polyester resins	(C_14_H_22_O_6_)_n_	170–172	[69]
Silicone polymers	(SiO_2_)_n_	1414	[70]
Ethylene-vinyl acetate (EVA) copolymers	(C_2_H_4_)_n_(C_4_H_6_O_2_)_m_	90	[71]
Polyvinyl alcohol (PVA)	(C_2_H_4_O)_n_	200	[72]
Ethylene propylene diene monomer (EPDM)	(C_8_H_16_)_n_	100–160	[73]
Polyvinyl acetate (PVAc)	(C_4_H_6_O_2_)_n_	60	[74]
Polycarbonate (PC)	(C_16_H_14_O_3_)_n_	220–230	[75]
Acrylonitrile butadiene styrene (ABS)	(C_8_H_8_·C_4_H_6_·C_3_H_3_N)_n_	105	[76]
Polyimides	(C_22_H_10_O_4_)_n_	360	[77]
Polyamide (Nylon)	(C_12_H_22_N_2_O_2_)_n_	220	[78]
Phenolic resins	C_8_H_6_O_2_	90–150	[79]
Urea-formaldehyde resins	C_2_H_6_N_2_O_2_	130	[80]
Melamine-formaldehyde resins	C_4_H_8_N_6_O	354	[81]
Polylactic acid (PLA)	(C_3_H_4_O_2_)n	150–160	[82]
Polyhydroxy alkanoates (PHA)	(C_6_H_10_O_2_)_n_	170	[83]
Starch-based polymers	(C_6_H_10_O_5_)_n_	200–220	[84]
Cellulose acetate	(C_10_H_16_O_8_)_n_	230	[85]
Lignin-based polymers	Varies	108–150	[86]

**Table 3 polymers-16-02045-t003:** Classification of UL-94 V testing.

UL-94 Vo	Each specimen must have the first flame (t_1_) and the second flame (t_2_) less than 10 s. The total time for the first and second flames (t_1_ + t_2_) across all five specimens must be less than 50 s. Additionally, the second and third flames (t_2_ + t_3_) must be less than 30 s for each specimen. There should be no after-flame or afterglow up to the holding clamp, and no burning drops are allowed.
UL-94 V1	Each specimen must have t_1_ and t_2_ less than 30 s. The total time across all five specimens must be less than 250 s. Additionally, the t_2_ and t_3_ flames (t_2_ + t_3_) must be less than 60 s for each specimen. There should be no after-flame or afterglow up to the holding clamp, and no burning drops are allowed.
UL-94 V2	Each specimen must have t_1_ and t_2_ less than 30 s. The total time (t_1_ + t_2_) across all five specimens must be less than 250 s. Additionally, the second and third flames (t_2_ + t_3_) must be less than 60 s for each specimen. There should be no after-flame or afterglow up to the holding clamp, but burning drops are allowed.

**Table 4 polymers-16-02045-t004:** Summary of some eco-friendly flame retardants from the above literature review.

Study	Flame Retardant	Key Results	Thermal Properties	Flammability Test Results	Ref.
Study on TA-based composites	Tannic acid (TA)	Time to failure: 15–27 min	Peak HRR: 211 vs. 108 kW/m^2^; total HRR: 37.2 vs. 24.4 MJ/m^2^	Lower fire growth rates: 2.43 vs. 1.27 kW/m^2^s^−1^	[120]
Study on biochar-furfurylated wood	Furfuryl alcohol and biochar	Enhanced thermal stability and reduced flammability up to 70%	Decreased effective heat of combustion; higher char residue	Higher mass loss at low temperatures	[121]
Study on mineral fillers	Aluminum/magnesium hydroxide and magnesium carbonate	Endothermic decomposition, increased heat capacity, and reduced flammability by up to 70%	Quantified fire-retardant effects	Improved LOI, UL-94, and cone calorimeter results	[123]
Study on bio-composites with lignin	Lignin with P, N, and Cu elements	Reduced heat release rate, total heat release, and smoke production	Increased char residues	Enhanced flame retardancy	[124]
Study on MPUC flame retardant	Carboxymethylation alkali lignin, phytic acid, and melamine-urea-glyoxal resin	Total heat release reduction: 56.8%; total smoke production decrease: 92.3%	LOI: 23.6% to 41.5%	Passed UL-94 V-1 rating	[126]
Study on CS-GO-APP coating	Chitosan, graphene oxide, and ammonium polyphosphate	LOI: 22 to 42; HRR decrease: 105.50 to 57.51 kW/m^2^; THR decrease: 62.43 to 34.31 MJ/m^2^	Decreased initial and maximum thermal decomposition temperature	Excellent durability in water resistance and abrasion tests	[122]
This study introduces a PVA composite enhanced with graphene oxide and phytic acid	Graphene oxide and phytic acid	Achieved exceptional flame retardancy	pHRR reduction of 88.6%; THR reduction of 66.5%	Maintained structural integrity for over 2400 s	[72]
Study on polycarbonate (PC) hybridization in wood flour/high-density polyethylene (HDPE) composites	Boric acid and polycarbonate	Improved fire retardancy and mechanical properties	Char residue rate increased by 6.7% at 28% PC content	Heat release rate reduced upon combustion	[75]
Phenolic resins based on two natural products, namely, lignin and tannins, were implemented as bio-based fireproofing coatings for wood	Lignin and tannins with inorganic nanoparticles	Reduced heat release during combustion, improved wood integrity, and delayed flame propagation	Improved thermal resistance with TGA	Comparable performance to top commercial coatings	[79]

## Data Availability

Not applicable.

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
