# Peer review of "Eco-Friendly Polymer Nanocomposite Coatings for Next-Generation Fire Retardants for Building Materials"

_polymers, 2024, doi:10.3390/polym16142045_

Round 1

Reviewer 1 Report

Comments and Suggestions for Authors

A manuscript entitled “Eco-Friendly Polymer Nanocomposite Coatings for Next-Generation Fire Retardants for Building Materials” is well written and structured by the author. The manuscript may be accepted after incorporating modifications in the manuscript. The changes required in the manuscript is as follow:

·       Author must compare the current work with already published review paper on same topic in second last paragraph of introduction section to highlight the significance/novelty of manuscript, over the already published manuscript.

·       Author must give proper citations in Table 1 and 2. What is the source of the information?

·       Author cite Figure 3c (line 188) before Figure 3 a and b (line 200). it looks very strange and confusing. It needs to rearrange figures in proper order as it is mentioned in text for proper flow of manuscript.

·       Author needs to add one summary table including quantitative details of the literature reviewed for flame retardants chemicals in section 5.

·       At the end, authors are suggested to included appropriate outlook figure for future roadmap of such materials.

·       Authors need to double check manuscript before submitting revision. There are some typo and grammatical errors.

Comments on the Quality of English Language

Minor editing of English language required

Author Response

A manuscript entitled “Eco-Friendly Polymer Nanocomposite Coatings for Next-Generation Fire Retardants for Building Materials” is well written and structured by the author. The manuscript may be accepted after incorporating modifications in the manuscript. The changes required in the manuscript is as follow:

Thank you for your positive recommendations. We have responded to each of your comments and incorporated the required modifications in the manuscript.

Author must compare the current work with already published review paper on same topic in second last paragraph of introduction section to highlight the significance/novelty of manuscript, over the already published manuscript.

Response: Thank you for your valuable suggestion. In the revised manuscript, we have included a comparison with existing review papers in the second-last paragraph of the introduction section. This comparison highlights the novel aspects and contributions in advancing the field of eco-friendly polymer nanocomposite coatings for fire retardancy.

Author must give proper citations in Table 1 and 2. What is the source of the information?

Response: Thank you for your comment. We have provided proper citations for each fire retardant chemical and polymer listed in Table 1 and Table 2.

Author cite Figure 3c (line 188) before Figure 3 a and b (line 200). it looks very strange and confusing. It needs to rearrange figures in proper order as it is mentioned in text for proper flow of manuscript.

Response: Thank you for pointing that out. We have now rearranged the figures in the proper order as mentioned in the text to ensure a coherent flow of the manuscript.

Author needs to add one summary table including quantitative details of the literature reviewed for flame retardants chemicals in section 5.

Response: Thank you for your suggestion. We have included a summary table (Table 4) that provides quantitative details of some eco-friendly fire retardant chemicals reviewed in Section 5.

At the end, authors are suggested to included appropriate outlook figure for future roadmap of such materials.

Response: Thank you for your suggestion. The manuscript thoroughly discussed the importance of current research trends, limitations, challenges, and opportunities related to eco-friendly fire retardants. Given the complexity and breadth of these topics, we initially found it challenging to create a concise and accurate roadmap figure. However, we have now included a probable future roadmap in Figure 9.

Authors need to double check manuscript before submitting revision. There are some typo and grammatical errors.

Response: Thank you for your feedback. We have carefully reviewed the manuscript and addressed all typos and grammatical errors.

Reviewer 2 Report

Comments and Suggestions for Authors

The article is interesting and relevant but needs revision before publication:

1. Treatment methods
- Subsection 2.1: the size of the numbers and the title of this subsection should be standardized.
- Impregnation, Coatings: Recommend to the authors to introduce subsections for these parts, i.e. make subsection 2.1.1. Impregnation and 2.1.2. Coatings.

2. Research on flame retardants chemicals
- The references in this section are not properly cited. Thus: lines 236-238 "For instance, Park et al. (2005) conducted a thermal analysis to examine the combustion characteristics of fire-retardant treated wood." it is necessary to change on "For instance, Park et al. [55] conducted a thermal analysis to examine the combustion characteristics of fire-retardant treated wood." Request for the authors in this section to make similar changes.

3. Tables
- All tables are not designed correctly. Recommend to the authors to familiarize themselves with the requirements of the template for this journal and to design the tables in accordance with them. Pay attention to the color of the lines, their delimitation, as well as the design of the table captions.

4. Figures
- All the figures are not designed correctly. Recommend to the authors once again to familiarize with the requirements of the template for this journal and in accordance with them to formalize the figures and their captions. 
- Figure 7: a scale marker needs to be added to the SEM images. Now it is "not clearly" visible, please correct it.

5. References
- Pay attention to the authors that the reference list should be designed according to the requirements of the journal template. Pay attention to font size and line spacing, please make changes.

6. Other comments
- It is necessary to add a paragraph indent in the following places: lines: 94, 96, 117, 132, 145, 152, 165, 172, 181, 193, 205, 214, 234, 246, 263, 276, 288, 299, 313, 329, 338, 345, 360, 375, 383, 425, 441. 
- There are extra line spacing throughout the Manuscript that should be removed, so lines: 95-96, 131-132, 171-172, 180-181, 245-246, 262-263, 275-276, 287-288, 298-299, 312-313, 344-345, 374-375, 382-383.

Author Response

The article is interesting and relevant but needs revision before publication:

Response: Thank you for your feedback. We appreciate your positive remarks about the relevance and interest of our article. We have carefully revised the manuscript to address your comments and suggestions, ensuring it meets the required standards for publication.

  1. Treatment methods

- Subsection 2.1: the size of the numbers and the title of this subsection should be standardized.

- Impregnation, Coatings: Recommend to the authors to introduce subsections for these parts, i.e. make subsection 2.1.1. Impregnation and 2.1.2. Coatings.

Response: Thank you for your suggestions. We have standardized the size of the numbers and the title format in Subsection 2.1. Additionally, we have introduced subsections for Impregnation (Subsection 2.1.1) and Coatings (Subsection 2.1.2) as recommended.

  1. Research on flame retardants chemicals

- The references in this section are not properly cited. Thus: lines 236-238 "For instance, Park et al. (2005) conducted a thermal analysis to examine the combustion characteristics of fire-retardant treated wood." it is necessary to change on "For instance, Park et al. [55] conducted a thermal analysis to examine the combustion characteristics of fire-retardant treated wood." Request for the authors in this section to make similar changes.

Response: Thank you for your feedback. We have revised the references in the specified section to ensure proper citation format. Each reference now includes the appropriate citation style, such as "Park et al. [55]," as requested

  1. Tables

- All tables are not designed correctly. Recommend to the authors to familiarize themselves with the requirements of the template for this journal and to design the tables in accordance with them. Pay attention to the color of the lines, their delimitation, as well as the design of the table captions.

Response: Thank you for your feedback. We have reviewed and corrected all tables in accordance with the journal's template requirements. To align with the journal's guidelines, changes have been made to ensure proper design, including the color of lines, their delimitation, and the formatting of table captions.

  1. Figures

- All the figures are not designed correctly. Recommend to the authors once again to familiarize with the requirements of the template for this journal and in accordance with them to formalize the figures and their captions.

- Figure 7: a scale marker needs to be added to the SEM images. Now it is "not clearly" visible, please correct it.

Response: Thank you for your comments. We have revised all figures to adhere to the journal's template requirements, formalizing their design and captions accordingly. Regarding Figure 7, please note that these figures are sourced from published work and cannot be further modified. We appreciate your understanding in this matter.

  1. References

- Pay attention to the authors that the reference list should be designed according to the requirements of the journal template. Pay attention to font size and line spacing, please make changes.

Response: Thank you for your feedback. We have made the necessary corrections to ensure that the reference list adheres to the journal template's requirements. Changes have been made to the font size, line spacing, and overall formatting to meet the specified guidelines.

  1. Other comments

- It is necessary to add a paragraph indent in the following places: lines: 94, 96, 117, 132, 145, 152, 165, 172, 181, 193, 205, 214, 234, 246, 263, 276, 288, 299, 313, 329, 338, 345, 360, 375, 383, 425, 441.

Response: Thank you for your feedback. As requested, we have added paragraph indents to the specified places in the manuscript.

- There are extra line spacing throughout the Manuscript that should be removed, so lines: 95-96, 131-132, 171-172, 180-181, 245-246, 262-263, 275-276, 287-288, 298-299, 312-313, 344-345, 374-375, 382-383.

Response: Thank you for your comments. We have addressed the extra line spacing issues throughout the manuscript as much as possible.

Thank you so much!

Reviewer 3 Report

Comments and Suggestions for Authors

The comprehensive review article by Kolya et. al. provides a timely overview of eco-friendly polymer nanocomposite coatings as next-generation fire retardants for building materials, particularly wood. The authors effectively contextualize the research within the broader push for carbon neutrality and sustainable construction practices. The review covers key aspects including synthesis methods, characterization techniques, flame retardancy mechanisms, and recent advancements in the field. However, to match the current research level in the field, this review article needs to be revise. Recommended corrections:

Questions for the authors:

1. You mention several eco-friendly nanofillers like graphene oxide and clay minerals. Could you provide a comparative analysis of their effectiveness and cost-efficiency? How might this guide future research priorities?

2. The review focuses primarily on wood as a substrate. How applicable are these nanocomposite coatings to other building materials? What modifications might be necessary for different substrates?

3. Given the importance of long-term durability in building materials, how do these eco-friendly nanocomposite coatings perform in accelerated aging tests? Are there concerns about their fire retardancy properties degrading over time?

4. You discuss various characterization techniques. Are there any emerging or novel characterization methods that could provide deeper insights into the fire retardancy mechanisms of these nanocomposites?

5. Update the reference list of the article to evaluate the recent literature in the field of nanocomposites 10.3390/jcs7100431 10.1007/s10967-024-09362-4

6. The environmental benefits of these materials are emphasized. Have there been any life cycle assessments comparing these eco-friendly nanocomposites to traditional fire retardants? What additional data would be valuable to fully understand their environmental impact?

7. While the review covers recent advancements, it would be helpful to have a clearer roadmap for future research. What specific challenges do you believe need to be addressed in the next 5-10 years to make these materials commercially viable on a large scale?

Author Response

The comprehensive review article by Kolya et. al. provides a timely overview of eco-friendly polymer nanocomposite coatings as next-generation fire retardants for building materials, particularly wood. The authors effectively contextualize the research within the broader push for carbon neutrality and sustainable construction practices. The review covers key aspects including synthesis methods, characterization techniques, flame retardancy mechanisms, and recent advancements in the field. However, to match the current research level in the field, this review article needs to be revise. Recommended corrections:

Thank you for evaluating our manuscript and your valuable suggestions for improvement. We appreciate your thorough review and are grateful for the opportunity to enhance the quality of our work.

Questions for the authors:

1. You mention several eco-friendly nanofillers like graphene oxide and clay minerals. Could you provide a comparative analysis of their effectiveness and cost-efficiency? How might this guide future research priorities?

Response: Thank you for your inquiry. Estimating the cost-efficiency of eco-friendly nanofillers like graphene oxide and clay minerals can be challenging due to various factors such as production methods, market fluctuations, and application-specific requirements. However, biobased materials often promise to be more cost-effective and environmentally friendly than their traditional counterparts. Future research can explore these biobased alternatives' scalability and economic viability to guide sustainable development in fire retardant materials.

2. The review focuses primarily on wood as a substrate. How applicable are these nanocomposite coatings to other building materials? What modifications might be necessary for different substrates?

Response: Thank you for your clarification. As a review article focusing on wood substrates to support carbon neutrality and sustainable construction practices, we acknowledge the potential applicability of the discussed nanocomposite coatings to other building materials as well. While our emphasis remains on wood due to its renewable and environmentally friendly qualities, the coatings' effectiveness and necessary modifications for different substrates could be an area of exploration for future studies aiming to extend their application beyond wood.

3. Given the importance of long-term durability in building materials, how do these eco-friendly nanocomposite coatings perform in accelerated aging tests? Are there concerns about their fire retardancy properties degrading over time?

Response: Thank you for raising this important point. In our review, we emphasize the need for assessing the long-term durability of eco-friendly nanocomposite coatings, including their performance in accelerated aging tests. While specific data on accelerated aging tests were not within the scope of our review, it is crucial to investigate how these coatings maintain their fire retardancy properties over extended periods. Future research should focus on conducting rigorous durability assessments to ensure that these coatings meet sustainability goals while maintaining effective fire retardancy throughout their intended lifespan in building applications.

4. You discuss various characterization techniques. Are there any emerging or novel characterization methods that could provide deeper insights into the fire retardancy mechanisms of these nanocomposites?

Response: Thank you for your inquiry. We have comprehensively covered the characterization techniques commonly used in the field, including thermal analysis, microscopy, spectroscopy, X-ray diffraction, and mechanical testing methods. While there are always advancements in characterization techniques, particularly in nanocomposite research, our review focused on established methods that provide robust insights into the fire retardancy mechanisms of these materials. If emerging or novel techniques exist in this area, we look forward to future research highlighting their application and contributions to understanding fire retardancy mechanisms in nanocomposites.

5. Update the reference list of the article to evaluate the recent literature in the field of nanocomposites 10.3390/jcs7100431 10.1007/s10967-024-09362-4

Response: Thank you for your suggestion. We have updated the article's reference list to include the recent literature you mentioned under reference number 94.

6. The environmental benefits of these materials are emphasized. Have there been any life cycle assessments comparing these eco-friendly nanocomposites to traditional fire retardants? What additional data would be valuable to fully understand their environmental impact?

Response: Thank you for your query. While our review emphasizes the environmental benefits of eco-friendly nanocomposites, specific life cycle assessments comparing these materials to traditional fire retardants were not within the scope of our review. Additional data from comprehensive life cycle assessments would be valuable in fully understanding their environmental impact. These assessments could include factors such as resource consumption, energy use, emissions, and end-of-life considerations, providing a holistic view of the ecological footprint of eco-friendly nanocomposites compared to traditional alternatives. Future research should prioritize conducting such assessments to support informed decision-making in sustainable material development.

7. While the review covers recent advancements, it would be helpful to have a clearer roadmap for future research. What specific challenges do you believe need to be addressed in the next 5-10 years to make these materials commercially viable on a large scale?

Response: Thank you for your insightful comment. We appreciate the need for a more precise roadmap for future research in developing and commercializing eco-friendly flame retardant materials. In response to your suggestion, we have illustrated a probable future roadmap in Figure 9.

Thank you so much!

Round 2

Reviewer 2 Report

Comments and Suggestions for Authors

Thank you to the authors for a job well done. After the finalisation, the Manuscript material looks more logical and structured. Recommend this material for publication in the journal MDPI "Polymers" in present form.